# Genome-Wide Analysis of Lipoxygenase (LOX) Genes in Angiosperms

**DOI:** 10.3390/plants12020398

**Published:** 2023-01-14

**Authors:** Paula Oliveira Camargo, Natália Fermino Calzado, Ilara Gabriela Frasson Budzinski, Douglas Silva Domingues

**Affiliations:** 1Group of Genomics and Transcriptomes in Plants, Department of Biodiversity, Institute of Biosciences, São Paulo State University, UNESP, Rio Claro 13506-900, SP, Brazil; 2Departamento de Genética, Escola Superior de Agricultura “Luiz de Queiroz”, University of São Paulo, USP, Piracicaba 13418-900, SP, Brazil

**Keywords:** lipoxygenase gene family, angiosperms, purifying selection

## Abstract

Lipoxygenases (LOXs) are enzymes that catalyze the addition of an oxygen molecule to unsaturated fatty acids, thus forming hydroperoxides. In plants, these enzymes are encoded by a multigene family found in several organs with varying activity patterns, by which they are classified as LOX9 or LOX13. They are involved in several physiological functions, such as growth, fruit development, and plant defense. Despite several studies on genes of the LOX family in plants, most studies are restricted to a single species or a few closely related species. This study aimed to analyze the diversity, evolution, and expression of LOX genes in angiosperm species. We identified 247 LOX genes among 23 species of angiosperms and basal plants. Phylogenetic analyses identified clades supporting LOX13 and two main clades for LOX9: LOX9_A and LOX9_B. Eudicot species such as *Tarenaya hassleriana*, *Capsella rubella*, and *Arabidopsis thaliana* did not present LOX9_B genes; however, LOX9_B was present in all monocots used in this study. We identified that there were potential new subcellular localization patterns and conserved residues of oxidation for LOX9 and LOX13 yet unexplored. In summary, our study provides a basis for the further functional and evolutionary study of lipoxygenases in angiosperms.

## 1. Introduction

Lipoxygenases (LOXs; EC 1.13.11.12) are enzymes belonging to the class of oxidoreductases that catalyze the addition of an oxygen molecule to unsaturated fatty acids, thus forming hydroperoxides that decompose into short-chain acids, aldehydes, and ketones. The most common plant fatty acids broken down by LOXs are linoleic and linolenic acids. LOXs are widely present in living organisms, occurring in bacteria, fungi, animals, and plants [1].

In plants, LOXs are found in several organs in varying concentrations; they are involved in several physiological functions including growth and development, vegetative reserve, senescence, resistance to insects and pathogens, seed germination, and as precursors of hormones and volatile substances [2]. It is known that lipoxygenase proteins effectively participate in the biosynthesis of the plant hormone jasmonate. Therefore, several physiological functions in plants depend on the association of these enzymes with this hormone. In tobacco plants, lipoxygenases are associated with responses involved in plant defense and resistance to stress through their regulatory elements such as methyl jasmonate (MeJA). Lipoxygenases are also involved, through MeJA biosynthesis, in metabolic pathways that regulate the transcription of the leaf senescence process, a fact observed in experiments carried out with the model species *Arabidopsis thaliana*. In *Cucurbita pepo*, the hormone jasmonate, synthesized by lox3a, controls petal elongation and flowering opening as well as fruit abortion in the absence of fertilization [3,4,5].

In higher plants, LOX enzymes can produce fatty acid hydroperoxides through two pathways known as the LOX pathways. The hydroperoxides formed are reactive molecules that can be mobilized in higher plants by enzymatic complexes involving enzymes such as hydroperoxide cyclase and hydroperoxide lyase. The latter, in turn, produces six-carbon compounds such as trans-2-hexenal, which is a characteristic component of fruit flavor and odor. Twelve-carbon compounds can also be produced by this enzyme, such as thaumatin, which is involved in signaling and cell division processes in response to plant injuries [6]. To date, LOXs have been classified according to their oxidation position of polyunsaturated fatty acids—LOX9 and LOX13 are responsible for the oxygenation of linoleic acid at carbons 9 and 13, respectively—or based on their cellular location—LOX type I was found in the cytoplasm, and LOX type II in the organelle-targeting signal peptides [7]. *Arabidopsis thaliana*, a reference plant for the evolutionary analysis presented in this study, contains six LOX genes, of which two are of the LOX9 type and four are of the LOX13 type [8].

So far, evolutionary studies on LOX genes in plants are restricted to a single species or a few closely related species [7,9,10]. Given this context, more detailed phylogenetic analyses were performed in this study using LOX members from 23 angiosperm plant species to comprehensively assess the relationships between plants and LOX enzymes.

## 2. Results

A total of 247 LOX genes were found among 23 plant species: *Arabidopsis thaliana*, *Citrus sinensis*, *Capsella rubella*, *Gossypium raimondii*, *Tarenaya hassleriana*, *Prunus persica*, *Eucalyptus grandis*, *Ricinus communis*, *Cucumis sativus*, *Capsicum annuum*, *Utricularia gibba*, *Daucus carota*, *Coffea canephora*, *Brachypodium distachyon*, *Setaria italica*, *Populus trichocarpa*, *Oryza sativa* ssp. *japonica*, *Musa acuminata*, *Sorghum bicolor*, *Picea abies*, *Marchantia polymorpha*, *Amborella trichopoda*, and *Chlamydomonas reinhardtii* (Appendix A).

In eudicots, the number of LOX genes varied between two (*Utricularia gibba*) and twenty (*Populus trichocarpa*), with an average number of genes of 11.29. In monocots, the number of genes varied between 10 (*Brachypodium distachyon*) and 16 (*Musa acuminata*); the average was 12 genes. In basal plants, the number of LOX genes varied between one (*Chlamydomonas reinhardtii*) and sixteen (*Marchantia polymorpha*), and the mean number of genes was 7.25 (Figure 1).

The evolutionary tree was constructed based on amino acid sequence alignments. The LOX genes were divided into three groups with bootstrap support above 90%. Therefore, according to our data, we proposed a new nomenclature of the clades as follows: LOX13 group, LOX9_A (previously called LOX9), and LOX9_B (Figure 2).

We found inconsistencies within the two groups of LOX13, which until now were classified as LOX13 type I and LOX 13 type II. Although LOX13 type II presented a signal peptide for targeting organelles, LOX13 type I could also present a signal peptide for signaling in organelles. Thus, it was possible to infer that cellular localization was not the main and only mode used to classify LOX13 proteins. The LOX13 type II genes showed two distinct subclades for division into monocots and eudicots. However, the LOX13 type I genes showed two subdivisions for eudicots and one for monocots, indicating a more complex evolution of the type I LOX13s. We also observed that the LOX9 clade had two distinct main sub-clades. There was a highly supported phylogenetic sub-clade, this being a sub-clade supported by external groups (*Amborella trichopoda* and *Picea abies*) with a distinct division between monocots and dicots; however, we found a regular distribution among the species of angiosperms, with an expansion of LOX genes in *Gossypium raimondii*. All the monocots used in this study had at least one representative of LOX9 in this sub-clade, but the same was not observed for the eudicots since not all the species had at least one representative of this sub-clade. The only angiosperm species that presented at least one copy of the LOX gene for this sub-clade were *Prunus persica*, *Populus trichocarpa*, *Ricinus communis*, *Eucalyptus grandis*, *Citrus sinensis*, and *Gossypium raimondii*. In the Brassicaceae species, including the model *Arabidopsis thaliana*, representative LOX genes for this putative new clade were detected.

We identified that representatives of the LOX9_B subgroup had, in one of their oxidation domains, a specific site for the conservation of the amino acids leucine (L) or methionine (M) (Figure 3), rather than valine (V), observed in all the members of the LOX9_A group.

LOX proteins were also analyzed based on their cellular location. LOX9_A and LOX9_B were 100% identified as cytoplasmic proteins. LOX13 type I (previous nomenclature) proteins were identified as cytoplasmic proteins (42%), chloroplasts (53%), or proteins present in another cellular compartment (5%). This same classification was observed for the LOX13 type II (previous nomenclature) proteins, such as cytoplasmic proteins (58%), chloroplasts (40%), or proteins present in other cellular compartments (2%) (Figure 4). Details are also available at https://doi.org/10.5281/zenodo.7374887 (accessed on 3 January 2022).

An analysis of the individual selection profile of each amino acid, as well as dN and dS substitution of the LOX genes in the eudicot and monocot species, was performed to verify the possibility of different evolutionary pressures. The clades LOX9_A (eudicotyledon and monocotyledon), LOX13 (A) (monocotyledon), and LOX13 (B) (eudicotyledon and monocotyledon) showed diversifying selection with a dN/dS value greater than 1. The clades LOX9_B (eudicotyledon and monocotyledon) and LOX13 (B) (eudicotyledonous) showed a purifying selection with a dN/dS value less than 1 (Figure 5).

Therefore, the selection models FEL, FUBAR, MEME, and SLAC were grouped in Venn diagrams (Figure 6 and Figure 7).

For the LOX9_A eudicots, eleven negative positions were common for the four models analyzed (FEL, FUBAR, MEME, and SLAC), and one positive position was common for MEME and FEL. In the LOX9_A monocots, the largest number of negative positions (11) was shared between the MEME and SLAC models, and the largest number of positive positions (16) was shared between the FEL, FUBAR, and MEME models. For LOX9_B, nine negative sites were shared between the SLAC and FEL models and forty-two positive sites were shared between the FUBAR and FEL models. In the LOX13 eudicots, the largest number of negative positions (68) was shared between the FUBAR and FEL models, and the largest number of positive positions (2) was shared between the SLAC and MEME models. For the LOX13 monocots, the FEL and MEME models shared the largest number of negative positions (8), and the FUBAR and FEL models shared 106 positive positions. Finally, positive selections detected with at least two different methods and moderately supported positive selections with only one method were categorized as strongly supported.

Public RNA-seq data were used to understand the LOX gene expression profiles in the angiosperms. Five plant species, including three eudicots (*Gossypium raimondii*, *Prunus persica*, and *Ricinus communis*) and two monocots (*Brachypodium distachyon* and *Sorghum bicolor*) were chosen based on the available literature. LOX genes were grouped into heatmaps according to their function: #LOX9_A, +LOX9_B, and *LOX13 (Figure 8).

To better understand the similarities between the LOX gene sequences generated in this study, the structural positions—exons and introns—were obtained for each identified clade (Appendix A). The maximum numbers of exons and introns found were ten and nine, respectively, as shown in DCAR_027194. Moreover, the minimum numbers of exons and introns identified were four and three, respectively, as shown in LOC_Os03g49380. The smallest LOX gene occurred in Cc02_g33800 and Cc02_g33320 (≅5 Kb), and the largest occurred in THA.LOC104807899 (≅29 Kb).

## 3. Discussion

The present study aimed to determine the number of LOX genes in several plant species, expanding the analyses to species that had never been studied, and to confirm the number of LOX genes, thus updating information regarding previously studied angiosperm species. The number of LOX genes in *Arabidopsis thaliana*, *Brachypodium distachyon*, and *Populus trichocarpa* identified in this study corroborated the number of LOX genes identified in previous studies [8,11,12]. However, we identified annotation errors for the LOX genes for some species. Shaban [7] identified fourteen LOX genes in *Gossypium raimondii*; in addition to these, we identified the presence of four additional LOX genes (Gorai.004G059500, Gorai.004G059900, Gorai.004G060100, and Gorai.004G059700); Gorai.004G092100, initially assigned as a LOX gene, was not included considering our parameters, as it did not present the domain IPR001024 (PLAT/LH2) and because its protein did not have a molecular weight of 90–110 kDa [13], thus having, in this sense, a high probability of being a pseudogene.

We identified 14 LOX genes in *Prunus persica*; in previous studies, the LOX copy number for this species ranged between 16 [14] and 12 [2]. The LOX genes ppa002308, ppa001112, and ppa001082 identified by Li [14] were grouped in the same branch in the present evolutionary tree, which led to the hypothesis that these genes are the result of alternative splicing. Using search tools (Blastn), we identified these three sequences as a single gene, coded as Prupe.047800 (PLAZAv.4 code) [15], so the sequences grouped in the previous study [14] may have resulted from alternative processing in *Prunus persica*. Studies on the evolution and regulation of the genes of the LOX family from alternative splicing processes have shown that alternative transcripts are regulated according to the stress variation to which a particular plant is subjected. This way, competitive or compensatory regulation mechanisms between isoforms arise [12].

A total of fourteen LOX genes have been described in *Oryza sativa* ssp. *japonica* [8]; however, in our study, we identified 11 LOX genes. LOC_Os12g37320 (55.29 kDa), LOC_Os02g19790 (50.74 kDa), and LOC_Os06g04420 (14.16 kDa) were not considered genes belonging to the LOX family as they did not present the domains IPR001024 (PLAT/LH2), IPR013819 (LOX, C-terminal), IPR001246 (LOX, plant), and IPR000907 (LOX), and these proteins were not of the average molecular weight (90–110 kDa) for the family. In *Capsicum annuum*, our study identified ten LOX genes, whereas Sarde [10] identified eight LOX genes for this species. The Capana03g003 sequence (59.16 kDa) was not included in our data because it did not present the average molecular weight for the LOX proteins. We used the strategies of repredicting the exon–intron structures of this gene in order to check if this was a problem of gene prediction. However, even so, the Capana03g003 gene did not meet the pre-established criteria in our study to be considered a gene of the LOX family. Through comparison analysis (Blastp), Capana01g001574 and Capana01g001578 were considered the same gene, as they were encoded as CAN.G649.19 (PLAZAv.4 code) [15]. In our first analyses, the CAN.G532.31 gene was not included as a LOX gene, as it did not present the IPR001024 (PLAT/LH2) domain; however, it had the molecular weight of average LOX genes. Therefore, CAN.G532.31 (with 2463 base pairs, 820 amino acids, and a molecular weight protein of 92.01 kDa) was included in the LOX gene family for *Capsicum annuum* in order to follow the nomenclature of Sarde [10].

Although we identified 17 LOX genes for *Cucumis sativus*, the presence of 23 LOX genes for this same species has been described in a previous study [16]. Csa013924 (57.97 kDa), Csa010340 (65.49 kDa), Csa009893 (82.93 kDa), and Csa019335 (49.93 kDa) were not included as LOX genes since they lacked IPR001024 (PLAT/LH2) and because they did not have the average molecular weight of LOX proteins. The Csa022479 gene, with a molecular protein weight of 29.94 kdA, did not present the domains IPR001024 (PLAT/LH2) and IPR001246 (LOX, plant). Finally, through the comparison analysis (Blastp), Csa006735 and Csa006736 were considered to be the same gene, which, in our analyses, was encoded by Cucsa.091350 (code PLAZAv.4) [15].

We found that, for the 247 sequences used in the construction of the evolutionary tree, the number of LOX9_B genes was much smaller when compared to the numbers of LOX9_A and LOX13 genes [10,17]. The LOX9_B group was restricted to *Amborella trichopoda* (basal, one gene), *Musa acuminata* (monocot, one gene), *Setaria italica* (monocot, one gene), *Oryza sativa* ssp. *japonica* (monocot, one gene), *Sorghum bicolor* (monocot, one gene), *Brachypodium distachyon* (monocot, one gene), *Populus trichocarpa* (dicot, two genes), *Prunus persica* (eudicot, one gene), *Ricinus communis*, (eudicot, one gene), *Eucalyptus grandis* (eudicot, one gene), *Gossypium raimondii* (eudicot, five genes), and *Citrus sinensis* (eudicot, one gene). Therefore, the LOX9_B genes were distributed among the species, mainly in only one copy, except for *Gossypium raimondii*, which presented five copies of LOX genes.

The LOX9_B subclade has already been reported in *Glycine max* and was considered to be exclusive to soybeans [18]. However, according to our results, LOX9_B had a wider distribution in the angiosperms. Using *Glycine max* LOX9_B as a query in https://shoot.bio/ (accessed on 4 April 2022), we confirmed that this clade was widespread in angiosperms, despite its patchy distribution (Appendix A). One hypothesis raised was that the LOX9_B genes may have been lost in eudicots over time. Eudicot species such as *Tarenaya hassleriana*, *Capsella rubella*, and *Arabidopsis thaliana* did not present any LOX9_B genes. We suggested that this loss in eudicots may have resulted from duplication events that occurred during the diversification of the Brassicaceae family, as in the present study we found *Tarenaya hassleriana* to be the representative species of this family. It is estimated that around 31.8 to 42.8 million years ago, close to the emergence of Brassicaceae, there was a duplication event where new classes of glucosinolates (compounds related to plant chemical defense) emerged [19]. Thus, both the duplication of glucosinolate genes and the loss of LOX_B genes in the Brassicaceae may have been favored during this evolution.

Another factor that reinforced the idea that the representatives of the LOX9_B subclade constituted a new group, when compared to other species of angiosperms, was the differential presence of conserved amino acids in a specific domain of lipoxygenase. We observed in our study that the species representing the LOX9_B group had a specific site with the conservation of the amino acids leucine (L) or methionine (M) in one of their domains. Vogt [9] identified, in this same position and in some plant species, including *Arabidopsis thaliana*, the amino acid valine (V) as conserved for the LOX9 group and the amino acid phenylalanine (F) as conserved for the LOX13 group (Figure 3). So, the LOX9_B clade is new to the literature.

Another point highlighted in our work was the way of classifying LOX proteins. In plants, most of the LOXs reported so far belong to LOX13, which plays a crucial role in the synthesis of jasmonates [1]. The LOX13 pathway catalyzes the conversion of unsaturated fatty acids (PUFAs) such as linolenic acid and arachidonic acid to hydroperoxide octadecatrienoic acid (HPOT13), which is metabolized in the plant as the signaling compounds jasmonates and green-leaf volatile compounds (GLVs). In *Physcomitrella patens*, a moss species, we demonstrated that the LOX13 type II (LOX13) protein acted on a linolenic acid substrate, whereas another LOX (LOX9_B) protein acts on an arachidonic acid substrate [20].

Up until now, the classification of LOX proteins was based on their oxidation position or cellular location [7,10,17]. LOX9 and LOX13 have been reported to be responsible for the oxygenation of linoleic acid at carbons 9 and 13, respectively. Furthermore, LOX9 enzymes have highly similar sequences, and the sequences of LOX13 type II (LOX13) enzymes are only moderately similar and contain an N-terminal chloroplast signal peptide, whereas LOX13 type I (LOX_B) enzymes have highly similar sequences and lack a chloroplast signal peptide [13,21]. However, this form of classification (LOX9, LOX13 type I, and LOX13 type II) is not the most adequate for grouping LOX proteins, as it is known that some LOX enzymes can perform both carbon-9 and carbon-13 oxidation [22,23]. Lipoxygenase proteins can also be classified based on their cellular location—LOX type I was found in the cytoplasm and LOX type II in the organelle-targeting signal peptides [7]. However, given our results, it was identified that LOX13 proteins, both type I and type II, were cytoplasmic proteins, proteins present in chloroplasts, or proteins present in another cell compartment. Thus, subcellular localization is not the best way to classify LOX proteins.

Finally, according to the literature, all type I LOXs are also necessarily type 13. However, the type II LOX group has a mix of type 9 LOXs and type 13 LOXs, which can cause classification errors [7]. Therefore, the re-annotation of LOX genes in angiosperm families in non-model species—as was carried out in our study—was necessary to improve the phylogenetic resolution.

After an analysis of the individual selection profile of amino acids in LOX proteins, we observed that the clades LOX9_A (eudicotyledonous and monocotyledonous), LOX13 (A) (monocotyledonous), and LOX13 (B) (eudicotyledonous and monocotyledonous) showed diversifying selection, that is, a dN/dS value greater than 1, suggesting that genetic modifications in the LOX genes for these clades were positively fixed throughout their evolution. The clades LOX9_B (eudicotyledonous and monocotyledonous) and LOX13 (B) (eudicotyledonous) presented a purifying selection, that is, a dN/dS value less than 1, suggesting a conservation of the function of the LOX genes for these clades. Thus, differences in selection pressure between the eudicotyledonous and monocotyledonous groups were observed only among the LOX13 clade (A). A ratio of dN/dS > 1 indicates acceleration, with evolution based on positive gene selection, while a ratio of dN/dS = 1 indicates that the genes are under the influence of a neutral selection action, and when the ratio of dN/dS is less than 1, the selection is indicated as purifying [24,25].

To understand the LOX gene expression profiles in angiosperms, we used public RNA-seq data from five plant species: three eudicots—*Gossypium raimondii* [26], *Prunus persica* [27], and *Ricinus communis* [28], and two monocots—*Brachypodium distachyon* [29] and *Sorghum bicolor* [30] (Figure 8). In *Brachypodium distachyon,* it was possible to notice that LOX9_A (Bradi1g11680 and Bradi1g11670) a greater expression value followed by LOX13 (Bradi3g07000 and Bradi3g07010), and LOX9_B presented the lowest expression value when comparing the LOX groups. LOX13 (Bradi3g07000 and Bradi3g07010) showed a higher expression in the control plants when compared to the plants submitted to immersion. In *Gossypium raimondii*, when studying the data obtained for leaves and roots (48, 12, and 0 h), the gene LOX13 Gorai.006G087200 had the highest expression value, and this same gene presented a differential expression between leaves (highest expression value) and roots (smallest expression value). For *Prunus persica*, when studying the leaves (control and water stress) and roots (control and water stress), the genes LOX9_B (Prupe.8G189000.1) and LOX13 (Prupe.2G005800.1 and Prupe.4g047800.1) showed the lowest values of expression. The Prupe.1G011400.1, Prupe.6g324600.2, Prupe.6G324100.1, Prupe.6G324300.1, Prupe.3G039200.1, Prupe.2G005300.1, and Prupe.1G232400.1 genes showed higher expression values in the control plants for root and water stress. Furthermore, Prupe.2G005500.1 and Prupe.6G018700.1 showed higher expression values for leaf control and water stress. These results showed that, regardless of the plant condition—control or water stress—these genes were related to the control of specific tissues. In *Ricinus communis*, LOX13 (RCO.g.30152.000070) showed a higher expression value in terms of flower development, which was followed by seed germination. The LOX13 gene (RCO.g.29929.000202) showed a higher expression value for flower development, and the LOX13 genes RCO.g.30169.000166 and RCO.g.30169.000164 showed higher expression values for development of leaf. In *Sorghum bicolor* (when considering control and fluphenim treatment), we observed higher expression values not for the specific groups of LOX but for both conditions, i.e., control and treatment. Thus, Sobic. 003G385500.1, Sobic. 001G125900.1, Sobic. 001G125800.1, Sobic. 003G385900.1, and Sobic. 006G095600.1 had the highest expression values.

The analysis of the structure and organization of the LOX genes revealed that the number of introns and exons varied little within each identified clade. That is, the function of the LOX genes within these clades was probably the same, corroborating the groups formed in the evolutionary tree.

## 4. Materials and Methods

### 4.1. Identification and Annotation of LOX Family Genes

Genomic sequences of LOX genes were obtained in twenty-three representative angiosperm species (Table 1) with a total of thirteen dicots, six monocots, the basal angiosperm *Amborella trichopoda*, and three species as outgroups: a gymnosperm (*Picea abies*), a bryophyte (*Marchantia polymorpha*), and a green alga (*Chlamydomonas reinhardtii*) (data A1 and data A2 in Appendix B).

Genes were searched by BLAST using LOX proteins from *Arabidopsis thaliana* as queries in PLAZA 4.0 [15], in which sequences that obtained a score greater than 200 and an e-value less than e-50 were recovered. All genes obtained were later manually analyzed to confirm the presence of typical LOX domains. We considered as LOX genes those that simultaneously presented the following InterPro domains in their respective proteins: IPR001024 (PLAT/LH2), IPR013819 (lipoxygenase, C-terminal), IPR001246 (lipoxygenase, plant), and IPR000907 (lipoxygenase). We also obtained in PLAZA [15], using an InterPro domain search, all genes that satisfied these criteria and were not found by a BLAST search. Besides domain composition, we selected the genes whose encoded proteins had a molecular weight between 90 to 110 kilodaltons for further analysis [13]. In the cases of gene prediction errors, gene prediction was confirmed using the FGNESH tool implemented on the Softberry website (http://www.softberry.com/) (accessed on 23 February 2022).

### 4.2. Multiple Sequence Alignment and Phylogenetic Analysis

The coding sequences (CDS) in nucleotides were aligned with MUSCLE [31] and translated into an amino acid alignment in the translatorX tool (http://translatorx.co.uk/) (accessed on 25 April 2019). Amino acid alignments were used to trace the phylogenetic profile of LOX family members using the maximum likelihood method in MEGAX [25], 1000 bootstrap replicates [32], Poisson’s model and uniform rates for the option ‘rates among sites’, and gaps in the alignment were treated as ‘pairwise deletion’. After running the protein model tests implemented in MEGA, we chose LG + G + I + F [33] as the best matrix model of amino acid substitution for the phylogenetic analysis. To retrack the evolutionary relationships among the 23 plant species, an evolutionary tree was constructed using PhyloT [34] and was visualized and annotated with iTOL [35].

### 4.3. Determination of Gene Structures

Gene Structure Display Server v2.0 [36] was used with standard parameters to analyze the exon–intron structure of the LOX genes. Genomic and CDS sequences in FASTA format corresponding to the genes of all the 23 plant species were inserted to generate the gene structures.

### 4.4. Selection Pressure and Evolutionary Analysis

Non-synonymous (dN) and synonymous (dS) nucleotide substitutions of the LOX gene sequences were classified and used for the dN/dS ratio. The indices dN/dS = 1, dN/dS < 1, and dN/dS > 1 represented Darwinian neutral evolution, purifying selection, or positive selection, respectively. Individual dN/dS indices for each amino acid of the predicted proteins for each gene were determined using the statistical test suite available in MEGAX [25]. Four sets of paralogous LOX genes, LOX9_A dicots and monocots, LOX9_B dicots and monocots, and LOX13, which was subdivided into (A) dicots and (B) monocots, were analyzed to detect positive and negative selection signatures. The position of sites subjected to positive selection was predicted with FUBAR, SLAC, FEL, and MEME based on a threshold *p*-value < 0.05 (or a posterior probability > 0.95). All these tools were implemented using the Datamonkey 2.0 online platform (https://datamonkey.org)(accessed on 26 November 2019) [37]. Positive and negative positions in each model were compared and grouped in Venn diagrams using the Bioinformatics & Evolutionary Genomics platform (http://bioinformatics.psb.ugent.be/webtools/Venn/)(accessed on 5 December 2019). Sites that evolved under positive selection were categorized as strongly supported (i.e., detected with at least two different methods) or moderately supported (i.e., detected with only one method). The files used for this analysis are available at https://doi.org/10.5281/zenodo.7374887.)(accessed on 7 December 2019)

### 4.5. Analysis of LOX Gene Expression Profiles in Angiosperms 

To understand the LOX gene expression profiles in the angiosperms, we used public RNA-seq data from five plant species: three eudicots, i.e., *Gossypium raimondii* [26], *Prunus persica* [27], and *Ricinus communis* [28], and two monocots, i.e., *Brachypodium distachyon* [29] and *Sorghum bicolor* [30]. Heatmaps were constructed with RPKM values obtained using CLC Genomics Workbench (CLC Bio–http://www.clcbio.com) (accessed on 3 March 2020).

### 4.6. Investigation of Motif Sequences and Cellular Localization of LOX Genes

LOX motif sequences were aligned using MAFFT version 7 [38] with the default parameters. The LOX recognition motifs were identified based on previously known domains [9]. The subcellular locations of all the LOX protein identified were also predicted. For this, two websites were used: CELLO v.2.5: subCELular localization predictor [39] and targetP-2.0 [40] (Data A3 in Appendix B).

## 5. Conclusions

In summary, we performed a comprehensive analysis of the LOX genes in 23 species of angiosperms and basal plants. We suggested that the 247 LOX members found in this study should receive a new nomenclature: LOX9_A, LOX9_B, and LOX13. The cell locations and oxidation positions of LOX9 and LOX13 should not be the most significant factors for classifying LOX genes. The distribution of these genes in the eudicots may indicate the loss of LOX9_B genes during the diversification process of the Brassicaceae family. The rhythm for LOX gene duplication and deletion events over time was not the same between the eudicot, monocot, and basal species. The pattern of synonymous substitution in the eudicots was higher than in the monocots; however, this was not observed in the groups LOX9_B and LOX13. Finally, the LOX expression profiles showed differential expression responses in tissues such as leaves and roots and in developing endosperms and seeds as well as a differential expression of LOX genes in the species subjected to water stress.

## Figures and Tables

**Figure 1 plants-12-00398-f001:**
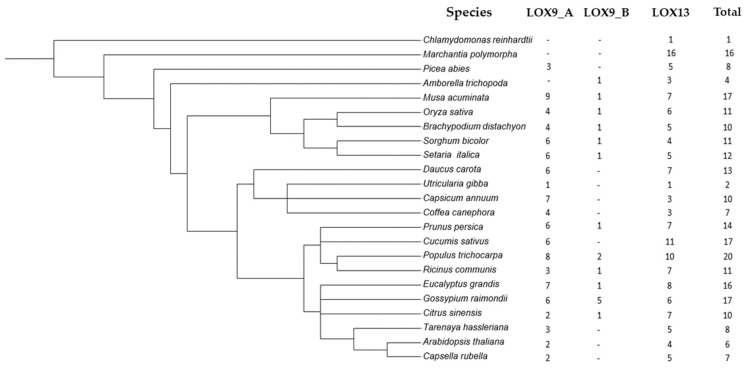
Number of LOX genes distributed among angiosperm groups. Fourteen species of eudicots, five species of monocots, and four basal species were analyzed.

**Figure 2 plants-12-00398-f002:**
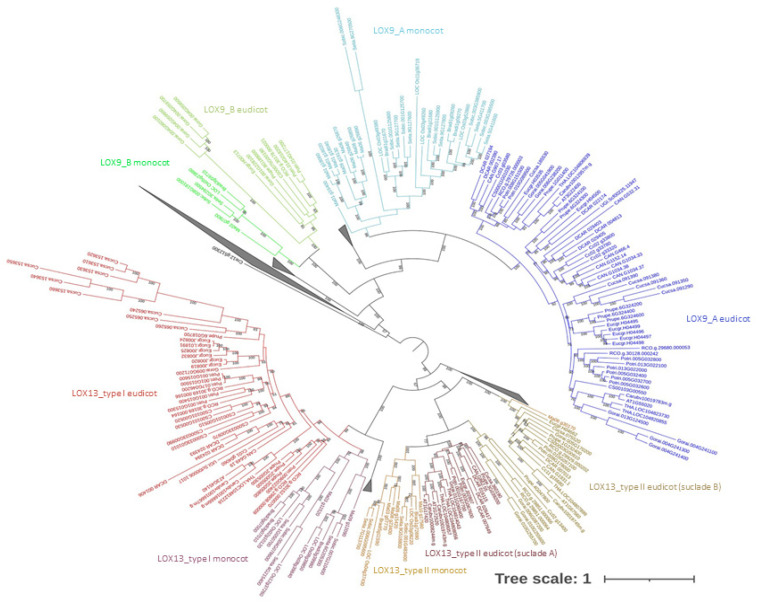
Evolutionary tree (maximum likelihood method, 1000 replicates per bootstrap) of lipoxygenases from 23 plant species. To facilitate visualization, we included the name of each clade with its corresponding color. Legend: eudicot LOX13_type I is in light-red, monocot LOX13_type I is in light-purple, monocot LOX13_type II is in light-brown, eudicot LOX13_type II (subclade A) is in dark-purple, and eudicot LOX13_type II (subclade B) is represented in dark-brown color. The LOX9_A group’s respective division among angiosperms is represented with the following colors: eudicot LOX9_A—dark blue and monocot LOX9_A—light blue. The LOX9_B group is represented in two shades of green: eudicot LOX9_B—dark green and monocot LOX9_B—light green.

**Figure 3 plants-12-00398-f003:**
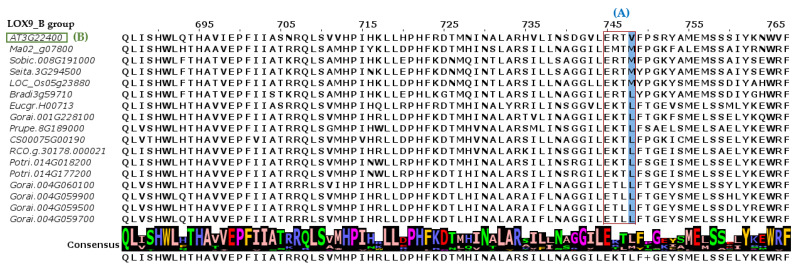
LOX9_B group and its pattern of amino acid conservation in motif sequences (**A**). This subclade had a specific conservation site for amino acids leucine (L) or methionine (M) (blue column). Diverging from the conservation pattern of LOX9 proteins in angiosperm species, including *Arabidopsis thaliana*, represented by AT3G22400 (**B**), a LOX9 gene, which had, in the same place (blue column), the amino acid valine (V), was conserved. We used MAFFT software version 7 for alignment and JALview for visualization.

**Figure 4 plants-12-00398-f004:**
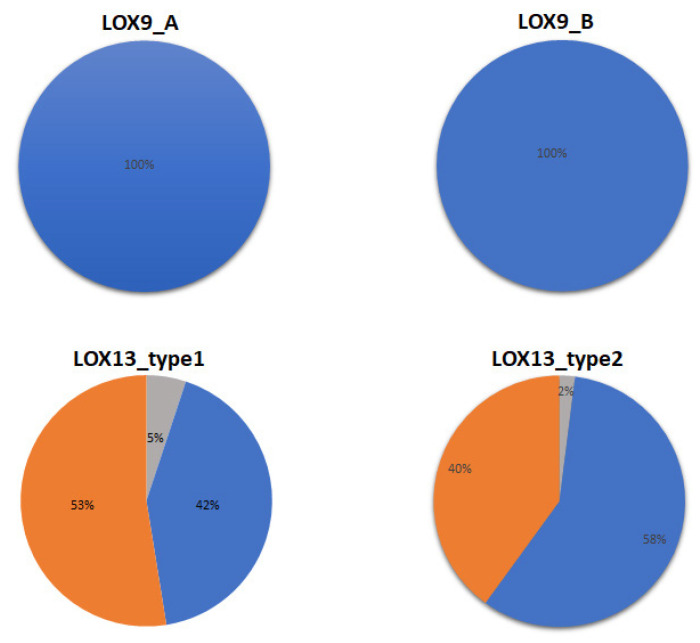
Subcellular localization of LOX proteins (in %). Proteins with cytoplasmic localization are represented in blue, proteins with signal peptide targeting chloroplasts are represented in orange, and proteins directed to other compartments are represented in gray.

**Figure 5 plants-12-00398-f005:**
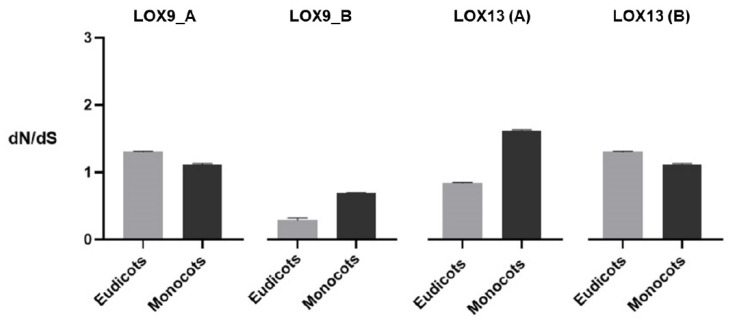
Comparison between the ratios of non-synonymous (dN) and synonymous (dS) substitutions between the LOX groups. Values obtained through the MEGAX program. Calculations performed on the Datamonkey platform (*p* < 0.05) using models of positive selections and dN/dS replacement ratios of LOX genes in eudicots and monocots.

**Figure 6 plants-12-00398-f006:**
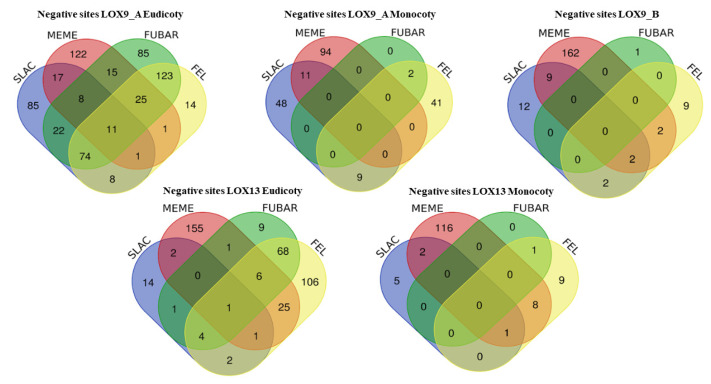
Venn diagrams. Comparison of negative sites between groups of lipoxygenases (LOX9_A eudicot, LOX9_A monocot, LOX9_B, LOX13 eudicot, and LOX13 monocot).

**Figure 7 plants-12-00398-f007:**
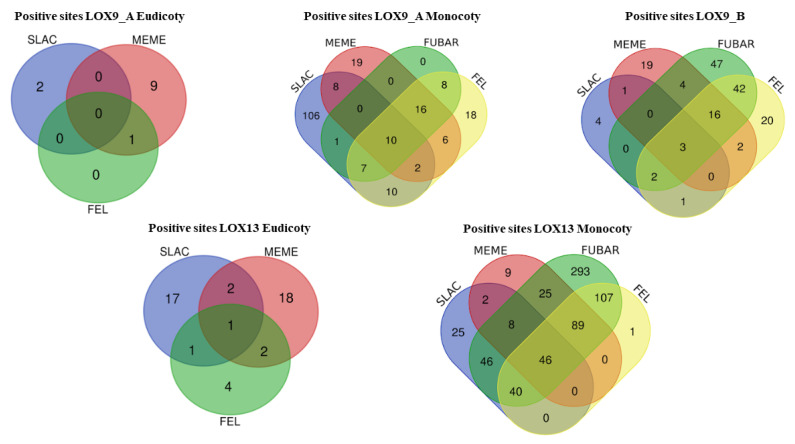
Venn diagram. Comparison of positive sites between groups of lipoxygenases (LOX9_A eudicot, LOX9_A monocot, LOX9_B, LOX13 eudicot, and LOX13 monocot).

**Figure 8 plants-12-00398-f008:**
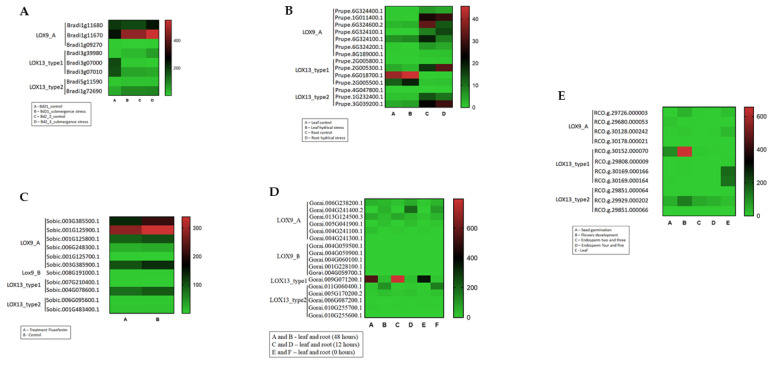
Transcription profile of the LOX gene family members in five angiosperm species (values in TPM—transcription per million). Symbols: LOX9_A, LOX9_B, LOX13_type 1 e LOX13_type 2. (**A**) *Brachypodim distachyon* (monocot.), control, and submersion stress. (**B**) *Prunus persica* (eudicot.), leaf (control and water stress) and root (control and water stress). (**C**) *Sorghum bicolor* (monocot.) control and treatment of Fluxophenim. (**D**) *Gossypium raimondii* (eudicot.), leaf and root (48, 12, and 0 h). (**E**) *Ricinus Communis* (eudicot.), seed germination, flower development, endosperm development II/III, endosperm development IV/V, and leaves. Data were obtained by the CLC Genomics Workbench program).

**Table 1 plants-12-00398-t001:** Species used for LOX analysis. Fourteen species of eudicots, five species of monocots, and four basal species were analyzed.

Species	LOX9-A	LOX9-B	LOX13	Total
*Chlamydomonas reinhardtii*	.		1	1
*Marchantia polymorpha*			16	16
*Picea abies*	3	.	5	8
*Amborella trichopoda*	.	1	3	4
*Musa acuminata*	9	1	7	17
*Setaria italica*	6	1	5	12
*Sorghum bicolor*	6	1	4	11
*Oryza sativa*	4	1	6	11
*Brachypodium distachyon*	4	1	5	10
*Daucus carota*	6		7	13
*Coffea canephora*	4		3	7
*Capsicum annuum*	7		3	10
*Utricularia gibba*	1		1	2
*Cucumis sativus*	6		11	17
*Prunus persica*	6	1	7	14
*Ricinus communis*	3	1	7	11
*Populus trichocarpa*	8	2	10	20
*Eucalyptus grandis*	7	1	8	16
*Gossypium raimondii*	6	5	6	17
*Citrus sinensis*	2	1	7	10
*Tarenaya hassleriana*	3		5	8
*Capsella rubella*	2		5	7
*Arabidopsis thaliana*	2		4	6

## Data Availability

Datasets containing protein subcellular localization results and FASTA files containing coding sequences (CDS) and protein sequences used for all phylogenetic analyses are available at https://doi.org/10.5281/zenodo.7374887 (accessed on 29 November 2022).

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
