# Peer review of "Genome-Wide Analysis of Lipoxygenase (LOX) Genes in Angiosperms"

_plants, 2023, doi:10.3390/plants12020398_

Round 1
Reviewer 1 Report
In this manuscript, the authors summarized and discussed the diversity, evolution, and expression of LOX genes in angiosperm species, which are involved in several physiological functions, such as growth, fruit development, and plant defense. On this basis, this manuscript further provides a basis for the functional and evolutional study of lipoxygenases in angiosperms. The manuscript is clearly stated. I would suggest accepting it after the following minor concerns are addressed.
Specific comments:
1. The manuscript is entitled “Lipoxygenase (lox) genes in angiosperms: A comparative genome-wide analysis”. However, most of the content was about Lipoxygenase while little refer to comparisons in angiosperms.
2. In the abstract, the authors wrote that this manuscript identified that there are potential new subcellular localization patterns and conserved residues of oxidation for LOX9 and LOX13. However, no further reasonable explanation was given in the manuscript.
3. In addition, the phylogenetic tree calculation must be described in more detail.
4. The physiological functions of LOXs in plants should be in depth discussed in the Introduction of this manuscript. Some relevant articles can be cited.
5. There are some grammar and format mistakes in the manuscript. (e.g., Reference 1,9 and 14)
Author Response
Reviewer 1
Reviewer(s)' Comments to Author:
“1. The manuscript is entitled “Lipoxygenase (lox) genes in angiosperms: A comparative genome-wide analysis”. However, most of the content was about Lipoxygenase while little refer to comparisons in angiosperms.”
Authors answer: 1. The title was modified for "Genome-wide analysis of Lipoxygenase (lox) genes in angiosperms”
“2. In the abstract, the authors wrote that this manuscript identified that there are potential new subcellular localization patterns and conserved residues of oxidation for LOX9 and LOX13. However, no further reasonable explanation was given in the manuscript.”
Authors answer: 2. We have added information about the subcellular localization of LOX proteins (in yellow in the text). Results: page 5 - 6th paragraph. Discussion: page 10 - 9th paragraph.
We also added the information about conserved residues of LOX9 and LOX13. Results: page 4 - 5th paragraph. Discussion: page 10 - 7th paragraph. Phrases corrected are highlighted in red color.
“3. In addition, the phylogenetic tree calculation must be described in more detail.”
Authors answer: 3. We added more details about the phylogenetic tree calculation. Materials and methods: page 12 - 3rd paragraph. Inclusions are highlighted in red color.
“4. The physiological functions of LOXs in plants should be in depth discussed in the Introduction of this manuscript. Some relevant articles can be cited.”
Authors answer: 4. The physiological functions of lipoxygenases in plants based on the literature were included. Introduction: page 1 - 2nd paragraph. Inclusions are highlighted in red color.
“5. There are some grammar and format mistakes in the manuscript. (e.g., Reference 1,9 and 14)”
Authors answer: 5. We corrected the errors and thank you for your observation.

Reviewer 2 Report
In the article entitled “Lipoxygenase (lox) genes in angiosperms: A comparative genome-wide analysis”, Paula et.al, investigated the diversity of LOX genes in angiosperm species. Their phylogenetic study identified LOX13 and LOX9 suggesting new nomenclature- LOX9_A and LOX9_B.
Their attempt is appreciable- gathering large amounts of genomic and transcriptomic data. The methodological part is very weak. For instance, the phylogenetic part of the study is not acceptable. The authors didn’t consider der Bayesian approach apart from the ML method to infer the cladding. Moreover, the text needs rigorous editing for grammatical errors and others.
I find it difficult to comment without line numbers in the text.
Please find my comments below:
Major Comments.
Abstract
What is evolutional study? I think the authors meant “evolutionary study”.
2. Results
Use the scientific nomenclature rules for the species names mentioned throughout the manuscript including ‘Figures’. For instance, in Figure 1, the rule is not followed in the phylogenetic tree.
Figure 3 legend is not clear. What does the box (highlighted) within the amino acid sequence alignment mean? Please clarify it in the figure legend. And also, clarify the left panel of Figure 3- representative species, eudicots, and monocots.
Figure 4 legend is not informative. What does the color indicate in the pie chart?
I am not clear what authors want to covey from the results obtained in Figure 4. It is clear from the text that dN/dS were calculated to evaluate the selection pressure. Did the authors detect any?
3. Discussion
Gorai.004G092100 was not assigned to the LOX gene category due to insufficient parameters set for LOX. How it was identified as a pseudogene here? Any pseudogene detection tool was used?
The discussion part is lacking proper connectivity and is not balanced. Please rewrite it in a focused way.
4. Materials and Methods
Which data was obtained first- the LOX gene or the whole-genome data of 23 plant species?
If the genomic data was obtained first, did you do an in-house BLAST or remote database search using PLAZA?
If it was performed with PLAZA, which version of the database was used for this study?
Which version of the InterPro database was used here? Was it an in-house or remote database?
4.6. Investigation of motif sequences and cellular localization of LOX genes
Why multiple approaches/methods were used for the prediction of the subcellular localization study?
Author Response
Reviewer 2
Reviewer(s)' Comments to Author:
- Abstract
“a) What is evolutional study? I think the authors meant “evolutionary study”.”
Authors answer: 1. a) That sentence was altered.
- Results
“a) Use the scientific nomenclature rules for the species names mentioned throughout the manuscript including ‘Figures’. For instance, in Figure 1, the rule is not followed in the phylogenetic tree.”
Authors answer: 2. a) Figure 1 was altered.
“b) Figure 3 legend is not clear. What does the box (highlighted) within the amino acid sequence alignment mean? Please clarify it in the figure legend. And also, clarify the left panel of Figure 3- representative species, eudicots, and monocots”
Authors answer: 2. b) Figure 3 was altered. The representative group was designated as LOX9_B and the subtitle was altered.
“c) Figure 4 legend is not informative. What does the color indicate in the pie chart?”
Authors answer: 2. c) The subtitle of figure 4 was altered. The cellular locations with their respective colors were informed.
“d) I am not clear what authors want to covey from the results obtained in Figure 4. It is clear from the text that dN/dS were calculated to evaluate the selection pressure. Did the authors detect any?”
Authors answer: 2. d) We added the results of the dN/dS calculations. Results: page 5 – 7th paragraph. The inclusion is highlighted in red.
- Discussion
“a) Gorai.004G092100 was not assigned to the LOX gene category due to insufficient parameters set for LOX. How it was identified as a pseudogene here? Any pseudogene detection tool was used?”
Authors answer: 3. a) Gorai.004G092100 was not considered as a LOX gene, because it does not have the IPR001024 domain (PLAT/LH2) and because its protein does not have a molecular weight between 90–110 kDa.
“b) The discussion part is lacking proper connectivity and is not balanced. Please rewrite it in a focused way.”
Authors answer: 3. b) The discussion was altered. Example: page 10 - 7th paragraph. The alteration is highlighted in red color.
- Materials and Methods
“a) Which data was obtained first- the LOX gene or the whole-genome data of 23 plant species?”
Authors answer: 4. a) First, we obtained the genes from Arabidopsis thaliana. Later the data of the 23 genomes. To better explain the process, we redid the phrase. Materials and methods: page 11 - 2nd paragraph. The alterations are highlighted in red color.
“b) If the genomic data was obtained first, did you do an in-house BLAST or remote database search using PLAZA?”
Authors answer: 4. b) We did a remote BLAST using the Arabidopsis thaliana genes.
“c) If it was performed with PLAZA, which version of the database was used for this study?”
Authors answer: 4. c) We use the PLAZA 4.0.
“d) Which version of the InterPro database was used here? Was it an in-house or remote database?”
Authors answer: 4. d) We use the interpro annotations already pre-available in PLAZA. PLAZA allows the individual verification of each gene, even indicating the associated interpro domains.
“e) 4.6. Investigation of motif sequences and cellular localization of LOX genes”
“f) Why multiple approaches/methods were used for the prediction of the subcellular localization study?”
Authors answer: 4. f) It is known that there are subcellular localization variations depending on the tool, as discussed in https://link.springer.com/article/10.1007/s11105-015-0898-2 . Other articles in the literature use this strategy (for example) https://journals.tubitak.gov.tr/agriculture/vol39/iss5/15/ and https://www.sciencedirect.com/science/article/pii/S0378111914006295).

Round 2
Reviewer 2 Report
In abstract section. Change to “functional and evolutionary study”
Restructure the sentence. Not only confusing, it is meaningless sentence. “Finally, after structural analyzes in the patterns of organization of exons and introns of LOX genes. Our study revealed that the number of exons and introns was little between clades. That is, the function of the genes in these groups is probably the same”
Probably use the sentence written in the previous version of the manuscript “The analysis of the structure and organization of the LOX genes revealed that the number of introns and exons varies little within each identified clade. That is, the function of the LOX genes in these clades is prob-ably the same, corroborating the groups formed in the phylogenetic tree.”
Author Response
Reviewer(s)' Comments to Author:
1. In abstract section. Change to “functional and evolutionary study”
2. Restructure the sentence. Not only confusing, it is meaningless sentence. “Finally, after structural analyzes in the patterns of organization of exons and introns of LOX genes. Our study revealed that the number of exons and introns was little between clades. That is, the function of the genes in these groups is probably the same”
Probably use the sentence written in the previous version of the manuscript “The analysis of the structure and organization of the LOX genes revealed that the number of introns and exons varies little within each identified clade. That is, the function of the LOX genes in these clades is prob-ably the same, corroborating the groups formed in the phylogenetic tree.”
Authors answer:
1. It was changed.
2. The sentence was substituted.
